# Metabolic Tolerance to Atmospheric Pressure of Two Freshwater Endemic Amphipods Mostly Inhabiting the Deep-Water Zone of the Ancient Lake Baikal

**DOI:** 10.3390/insects13070578

**Published:** 2022-06-24

**Authors:** Ekaterina Madyarova, Yulia Shirokova, Anton Gurkov, Polina Drozdova, Boris Baduev, Yulia Lubyaga, Zhanna Shatilina, Maria Vishnevskaya, Maxim Timofeyev

**Affiliations:** 1Institute of Biology, Irkutsk State University, 664025 Irkutsk, Russia; madyarovae@gmail.com (E.M.); yuliashirokova2501@gmail.com (Y.S.); a.n.gurkov@gmail.com (A.G.); drozdovapb@gmail.com (P.D.); baduevbk@gmail.com (B.B.); yuliya.a.lubyaga@gmail.com (Y.L.); zhshatilina@gmail.com (Z.S.); 2Baikal Research Centre, 664011 Irkutsk, Russia; 3Research Resource Center “Chromas”, Saint-Petersburg State University, 198504 Saint Petersburg, Russia; wishm@yandex.ru

**Keywords:** Amphipoda, antioxidant defense, Baikal, decompression, deep sea, deep water, energetic metabolism, eurybathic, lipid peroxidation, scavengers

## Abstract

**Simple Summary:**

Deep-water habitats are the largest ecosystem on the planet: over half of the Earth’s surface is covered with a water layer deeper than 200 m and remains poorly explored. Lake Baikal is the only freshwater body inhabited by animals adapted to the deep-water zone independently from their marine counterparts. Comparing these convergently evolved freshwater and marine animals is invaluable for revealing the basic mechanisms of adaptation to high hydrostatic pressure. However, laboratory experiments on deep-water organisms still usually require lifting them to the water’s surface and exposing them to potentially hazardous decompression, while endemics from Lake Baikal are poorly studied in this regard. Here, we compared metabolic reactions to such pressure decreases in two Baikal deep-water amphipods (shrimp-like crustaceans) from the genus *Ommatogammarus*: one species is known to tolerate pressures close to atmospheric levels, while the second was only observed at the pressures from 5 atm and above. We expected that the energy metabolism of the shallower-dwelling species would function better under the atmospheric pressure but found no substantial differences. Thus, despite some difference in long-term survival at atmospheric pressure, both species are suitable for laboratory studies as freshwater model objects adapted to large pressure variations.

**Abstract:**

Lake Baikal is the only freshwater reservoir inhabited by deep-water fauna, which originated mostly from shallow-water ancestors. *Ommatogammarus flavus* and *O. albinus* are endemic scavenger amphipods (Amphipoda, Crustacea) dwelling in wide depth ranges of the lake covering over 1300 m. *O. flavus* had been previously collected close to the surface, while *O. albinus* has never been found above the depth of 47 m. Since *O. albinus* is a promising model species for various research, here we tested whether *O. albinus* is less metabolically adapted to atmospheric pressure than *O. flavus*. We analyzed a number of energy-related traits (contents of glucose, glycogen and adenylates, as well as lactate dehydrogenase activity) and oxidative stress markers (activities of antioxidant enzymes and levels of lipid peroxidation products) after sampling from different depths and after both species’ acclimation to atmospheric pressure. The analyses were repeated in two independent sampling campaigns. We found no consistent signs of metabolic disturbances or oxidative stress in both species right after lifting. Despite *O. flavus* surviving slightly better in laboratory conditions, during long-term acclimation, both species showed comparable reactions without critical changes. Thus, the obtained data favor using *O. albinus* along with *O. flavus* for physiological research under laboratory conditions.

## 1. Introduction

The formal border of the deep sea or deep ocean is usually defined as >200 m, which makes the deep-water zone the largest habitat on the planet, covering more than half of earth’s surface [1,2]. Deep-water organisms play an inestimable role in the biosphere, and yet, relatively little is still known about their biodiversity, and even less is known about their physiology [3,4]. This is especially worrying since global climate change is now expected to threaten biodiversity not only in surface waters but in deeper ones as well [4,5].

The crucial factor in deep-water habitats is high hydrostatic pressure increasing by about 1 atm with each 10 m of water column. Elevated pressure at great depths influences the folding of proteins and their interaction with ligands, as well as the fluidity of cell membranes, but specific adaptations to these effects are rarely studied in detail, and especially little is known for vertically migrating animals [6]. Some reports demonstrate that increasing hydrostatic pressure can cause elevated production of reactive oxygen species (ROS) and a related increase in lipid peroxidation [6]. Overall, one can expect greater energy demand for cellular adaptation to drastic pressure changes. High pressure significantly complicates physiological experiments with deep-water organisms since transporting them en masse to the laboratory without at least a short-term decrease in atmospheric pressure is still challenging [7]. In these circumstances, the animals dwelling in a wide vertical range from close to the surface to the deep waters are of especial interest as potential model objects, which both can tolerate pressure decreases before laboratory experiments and already possess the adaptation mechanisms of deep-water species.

Deep-water organisms are also constantly exposed to low temperatures, darkness, and food deficiency [8,9]. In these energetically scarce conditions, we can expect a significant reduction in organisms’ adaptive potential to unexperienced factors such as higher temperature [10] or solar ultraviolet radiation. Tracing such reductions in the species that colonized deep-water zones at different times is of interest for better understanding the roles of different cellular adaptation mechanisms in general, but can also be important for predicting reactions of deep-water ecosystems to global climate change.

Within deep-water fauna, amphipods (Amphipoda, Crustacea) are the dominant marine scavengers [11]. The diversity of deep-water amphipods is extremely rich and includes representatives of large phylogenetic lineages [12], showcasing a parallel adaptation to extreme habitat conditions and feeding habits. For example, it was shown that in the Lysianassoidea superfamily, the most speciose group of deep-sea scavengers, the morphological characters traditionally used for taxonomy are in fact subject to convergent evolution [13]. Different representatives of this group independently shifted from opportunistic to obligate scavenger lifestyles twice during their evolution [14]. Many species are benthopelagic and are suggested to play an important role in the upward transfer of organic matter [11]. However, the overwhelming majority of the data concerning deep-water amphipods come from marine environments.

Lake Baikal, in the middle of Eastern Eurasia, is an outstanding freshwater reservoir, the only one possessing an oxygenated deep-water zone down to the maximal depth of over 1.6 km, and hence rich deep-water fauna [15]. The lake is inhabited by more than 2500 mostly endemic species [16], and a substantial part of this biodiversity is amphipods that colonized numerous ecological niches at all depths of Baikal [17]. Those amphipods descend from shallow-water species of Eurasia belonging to the Gammaroidea superfamily [18,19,20] and now include deep-water scavengers from several genera [17]. The genus *Ommatogammarus* is one of them and includes some benthopelagic species inhabiting wide ranges of depths, which may allow their use as model objects for investigating the adaptation mechanisms of Baikal deep-water amphipods and comparing them to marine counterparts. Intriguingly, their overall morphology is convergently similar to that of the marine scavengers of the Alicelloidea and Lysianassoidea superfamilies (compare Figure 1 below with photos in [21]).

*O. flavus* (Dybowsky, 1874) has been found from depths of 2.5 to over 1300 m [22] and should thus be sufficiently adapted to atmospheric pressure. Previously, we used this feature of *O. flavus* to compare its reaction to an increasing temperature with shallow-water amphipods and indeed observed higher sensitivity for the deep-water species [23]. *O. albinus* (Dybowsky, 1874) is known to migrate down to the maximal depth of the lake (over 1600 m), but this species was found no higher than 47 m depth [22,24]. Paler pigmentation and available sporadic sampling data suggest that *O. albinus* probably prefers greater depths than *O. flavus* and thus may be even more important as a deep-water model object. However, it is not clear how *O. albinus* reacts to pressure decreases down to 1 atm at the physiological level, since this species has never been observed at pressure lower than about 5 atm.

So, in this study we aimed to compare the reactions of *O. flavus* and *O. albinus* to pressure decreases and further acclimation under laboratory conditions. Significant pressure decreases as well as increases may cause malfunctioning of the electron transport chain in mitochondria, which in turn can lead to electron leakage and increases in ROS production with subsequent damage to the cell. Most ROS have a very short half-life, and hence their elevated production is usually determined by successive products of the peroxidation of cellular components such as lipids or by indirect proxy markers such as the activities of various antioxidant enzymes, which usually should balance ROS [25,26]. Another basic requirement to survive under stressful conditions other than preventing oxidative stress is maintaining a sufficient energy supply. The most effective extraction of energy from glucose is oxidative phosphorylation in mitochondria, in contrast to anaerobic glycolysis, which produces lactate and depletes glycogen reserves quicker. The energy status of an organism can be estimated via such markers as the content of free glucose and glucose in the form of glycogen, ratios between energy-reach ATP and its dephosphorylated counterparts, as well as the activity of lactate dehydrogenase [27,28,29]. Thus, here, we focused on these key parameters of energy metabolism, as well as several markers of oxidative stress to compare how cells of these amphipods, sampled from different depths, cope with atmospheric pressure.

## 2. Materials and Methods

### 2.1. Amphipod Sampling

The studied species *Ommatogammarus flavus* (Dybowsky, 1874; representative photo in Figure 1) and *O. albinus* (Dybowsky, 1874; Figure 1) are neither endangered nor protected, and all further procedures with them were approved by the Animal Subjects Research Committee of the Institute of Biology at Irkutsk State University (protocol #003). Endemic amphipods were sampled from under the ice surface of Lake Baikal near the Bolshie Koty settlement (Southern Baikal; 51°91′37″ N, 105°06′91″ E) in March 2015 (the 1st sampling campaign) and March 2016 (the 2nd sampling campaign) using deep-water traps deployed at different depths of the lake bottom (Figure 1): approximately 50, 100, 150, 200 and 300 m in 2015, and 100, 150, 200, 300, 500, 750 and 1000 m in 2016. Rotten fish in net bags was used as the bait to attract the scavenger amphipods to each trap, which had been prepared to prevent the animals from escaping and also to ensure water exchange for a good oxygen supply. The traps were deployed for 3–11 days, lifted to the ice surface (approximately 11 m per min), transferred to water chambers with the temperature in the range 3–4 °C and opened to separate all trapped amphipods by species as quick as possible. A necessary number of adult animals from each depth was fixed in liquid nitrogen during the separation by species, and the rest (if any) were brought alive to the laboratory for acclimation.

### 2.2. Laboratory Acclimation

Water temperatures were maintained in the range of about 3–4 °C as much as possible during all procedures with deep-water amphipods, since the temperature at great depths of Lake Baikal is known to vary in this range [30,31]. *O. flavus* and *O. albinus* were kept separately in well-aerated aquaria with Baikal water in the incubator MIR-254 (Sanyo, Tokyo, Japan). The incubators have a narrow window for observation, but we tried to maintain dark conditions inside (no more than approximately 20 lx during a bright day). Darkness could not be maintained only during water exchanges (illuminance could reach approximately 150 lx in such a case) performed every 2–3 days after overnight feeding of the animals with frozen Baikal whitefish. The illuminance was measured with the lux meter DT-8809A (CEM, Macao, China). The acclimation to atmospheric pressure at laboratory conditions lasted for 9–12 days for both species in 2015, and for 13–16 days for *O. albinus* and 18–27 days for *O. flavus* in 2016 (*O. flavus* was acclimated for a longer period of time due to an accidental temperature increase to 8 °C), after which the amphipods were fixed in liquid nitrogen. A detailed schedule of all procedures with the studied animals is presented in Appendix A (list “Schedule”).

### 2.3. Sample Processing

For each depth, we analyzed from 3 to 8 samples per species (see Appendix A for raw data). Each sample included 3–5 frozen adult individuals crushed under liquid nitrogen. The homogenate was divided into three portions for the following measurements: activities of antioxidant enzymes and lactate dehydrogenase; the level of lipid peroxidation products in neutral lipids and phospholipids; and the content of glucose, glycogen and adenosine phosphates. The enzymatic activities and content of lipid peroxidation products were measured in 3 analytical replicates for each sample. Concentrations of adenylates, glucose and glycogen were measured in 2 analytical replicates.

### 2.4. Enzymatic Activities

Activities of catalase (CAT; EC 1.11.1.6), total peroxidases (POD; EC 1.11.1.7), glutathione S-transferase (GST; EC 2.5.1.18) and lactate dehydrogenase (LDH; EC 1.1.1.27) were analyzed using spectrophotometric assays. The enzymes were extracted in 0.1 M sodium phosphate buffer (pH 6.5), centrifuged at 10,000× *g* for 3 min, and the supernatant was used for enzymatic assays. Catalase activity was measured using hydrogen peroxide as a substrate at 240 nm (pH 7) [32]. The total activity of cellular peroxidases was determined using guaiacol and hydrogen peroxide as substrates at 436 nm (pH 5) [33]. Glutathione S-transferase activity was measured using 1-chloro-2,4-dinitrobenzene and glutathione as substrates at 340 nm (pH 6.5) [34]. Lactate dehydrogenase activity was determined using a commercial “LDG-vital” kit (Vital Development, Saint Petersburg, Russia) at 340 nm (pH 7.5) according to the manufacturer’s instructions. The total protein content was determined according to Bradford [35] in order to normalize all enzymatic activities and express them in nkat mg^−1^ of protein.

### 2.5. Content of Lipid Peroxidation Products

The levels of lipid peroxidation products were measured according to the modified method of Khyshiktuev [36] as described by Vereshchagina [37] in neutral lipids (mostly storage lipids; the upper heptane fraction) and phospholipids (mostly cellular membranes; the lower isopropanol fraction). Briefly, the samples were homogenized in heptane–isopropanol (1:1); then, the volume of the mixture was brought up to 4.5 mL; 1 mL of distilled water was added, and the mixture was incubated for 30 min at 25 °C in order to let the fractions separate. The fractions were then centrifuged, and the supernatant was mixed with ethanol (1:3). The lipid peroxidation products were measured using spectrophotometric analysis at 232 nm for diene conjugates (DC), 278 nm for triene conjugates (TC) and 400 nm for Schiff bases (SB). The content of each product in each fraction was expressed in arbitrary units by normalizing them to the amount of isolated double bonds as measured at 220 nm.

### 2.6. Concentrations of Energy-Related Molecules

Crushed frozen tissues were additionally homogenized in 15 mM ethylenediaminetetraacetic acid mixed with 0.6 M HClO_4_ (1:9) and centrifuged at 15,000× *g* for 15 min at 4 °C to remove precipitated proteins. The supernatant was neutralized with 5 M potassium carbonate and centrifuged to remove the precipitated perchlorates. The glycogen content (from non-neutralized and non-centrifuged supernatant) was determined after its enzymatic hydrolysis to glucose with α-amyloglucosidase and calculated as the difference in tissue glucose levels before and after hydrolysis using the previously described methods [38,39]. The contents of glucose, ATP, ADP and AMP were measured spectrophotometrically using the NADH/NADPH-dependent enzymatic methods at 340 nm and normalized to the wet tissue weight [39,40,41,42]. The adenylate energy charge (AEC) was calculated as described by Atkinson [43] using the following equation:(1)$$\mathrm{Adenylate}\;\mathrm{energy}\;\mathrm{charge}=\frac{\left[\mathrm{ATP}\right]+0.5\left[\mathrm{ADP}\right]}{\left[\mathrm{ATP}\right]+\left[\mathrm{ADP}\right]+\left[\mathrm{AMP}\right]}$$
where [ATP], [ADP] and [AMP] are the concentrations of adenylates (μmol g^−1^ of tissue).

### 2.7. Data Analysis and Statistics

The statistical processing, analysis and data visualization were performed using R software [44] with the packages ggplot2 [45] and ggpubr [46]. In the case of boxplot data visualization, outlier data points are plotted individually (beyond the ends of the whiskers) according to the default algorithm of the used packages. Correlation with the depth of sampling was determined using the Pearson coefficient. The samples were always compared using the nonparametric Mann–Whitney U-test (implemented in the wilcox.test function) with the Holm correction for multiple comparisons. The whole set of used scripts can be found via the following link: https://github.com/drozdovapb/code_chunks/tree/master/Ommatogammarus_deep_water_biochemistry (accessed on 11 April 2022).

## 3. Results

### 3.1. Vertical Distribution of O. flavus and O. albinus

Deep-water scavenger amphipods were collected from the bottom of Lake Baikal in two sampling campaigns (Figure 1). During both samplings, we observed no dead individuals of the studied species in the traps. In the 1st sampling campaign, the set of depths was relatively restricted and included samples from approximately 50 to 300 m. As expected, *O. flavus* was more abundant in traps from smaller depths (peak at 100 m), and *O. albinus* became more numerous from 200 m, with its maximum at 300 m. Such results clearly indicate that we did not cover the main distribution range of *O. albinus* and probably even shifted the native distributions of both species by making them migrate to 4 traps densely located within the first 200 m.

In the 2nd sampling campaign, we tried to prevent such interference and installed the traps from about 100 to 1000 m in order to capture a more native distribution of both deep-water species (Figure 1). *O. flavus* was concentrated in the range of 100–200 m despite some individuals being found down to 750 m. *O. albinus* started to appear from 150 m and prevailed at 500 m, and some individuals of this species were observed down to 1000 m. Even though the second pattern probably reflects the preferred depths of the studied *Ommatogammarus* species much better, all obtained results clearly confirm the hypothesis that *O. albinus* generally prefers substantially greater depths than *O. flavus*.

### 3.2. Variability of Biochemical Traits with Sampling Depth

We analyzed a number of biochemical traits in the sampled amphipods that were frozen as soon as possible after being lifted from each depth. In particular, we measured the activities of three antioxidant enzymes (CAT, POD and GST), content of lipid peroxidation products (DC, TC and SB in neutral lipids and phospholipids) and levels of several metabolic parameters (activity of LDH, concentrations of glucose, glycogen, ATP, ADP and AMP). All obtained data are presented in Appendix A separately for each year and sampling depth.

Correlation analysis between the measured parameters and depth showed that the absolute value of the Pearson coefficient never reached 0.6 in any case. Moreover, no marker had the correlation coefficients consistently over 0.4 or less than −0.4 in both sampling campaigns. Thus, there was no apparent influence of the sampling depth on any of the studied biochemical traits. This conclusion allowed us to combine the data by depth (despite the depths being different across different species and years) for further interannual and interspecific comparisons.

### 3.3. Interannual and Species-Specific Differences in Biochemical Traits

Figure 2 and Figure 3 present the combined data for each species for the metabolic parameters and markers of oxidative stress, respectively. We found some statistically significant differences between the different sampling years. In the 2nd sampling campaign, the glycogen content of *O. albinus* was higher (~1.3 times difference at the median) than in the 1st sampling campaign, despite the glucose level being lower (2-fold difference at the median). The activity of antioxidant enzymes was higher in the second sampling campaign for both species except CAT of *O. albinus*, where the difference was not statistically significant. In the case of lipid peroxidation products, the only difference was found for SB in *O. flavus*, with a median higher in the first sampling campaign.

We found several statistically significant differences between the *Ommatogammarus* species in either sampling year. This was the case for glucose and glycogen (the first sampling campaign), ATP (the second sampling campaign), ADP (the first sampling campaign) and TC in neutral lipids (the second sampling campaign). However, consistent interspecific differences over both years were only observed for GST and DC in neutral lipids. In both cases, *O. flavus* demonstrated higher values than *O. albinus* (an approximately 2-fold median difference for GST and ~1.5 times for DC).

In order to reduce the possible influence of differences in sampling depths on these comparisons, we additionally subsetted the data for intersecting depths over 2 years and performed analogous comparisons for samples from 100–300 m, presented in Appendix A. In this version, there are interannual statistically significant differences for glycogen in *O. flavus* and ADP in *O. albinus* that are not found in Figure 2. Some differences became non-significant in Appendix A, such as the interannual difference for glycogen in *O. albinus*, but higher activities of GST and POD in the second sampling campaign stay statistically significant for both species. Similarly, the described consistent interspecies difference in GST activity and the level of DC in neutral lipids is still fully supported.

### 3.4. Survival during Laboratory Acclimation

We applied laboratory acclimation in order to compare the potential of the two *Ommatogammarus* species to adapt to atmospheric pressure for possible long-term experiments. Since a high number of animals was required for any measurements after the acclimation, it was only possible for some depths. For *O. flavus*: 100 and 150 m both in the 1st sampling campaign and the 2nd sampling campaign, 200 m in the 2nd sampling campaign and 300 m in the 1st sampling campaign. For *O. albinus*: 200 and 300 m in the 1st sampling campaign and 500 m in the 2nd sampling campaign. During the acclimation, we noticed that *O. albinus* molts much more often than *O. flavus* and also, unlike *O. flavus*, does not form amplexus despite both species being known to breed year-round [17]. Both can be markers of higher stress for *O. albinus*, but we cannot provide any quantitative data to corroborate this.

The cumulative mortality (Figure 4 and Appendix A, sheet “Mortality during acclimation”) of *O. flavus* collected from 100–300 m varied approximately from 1 to 10% after 9–27 days of acclimation and was positively correlated with sampling depth (the Pearson coefficient for two years combined was 0.75). In the case of *O. albinus*, from 200–500 m, the correlation with depth was very strong (Pearson coefficient ~1), but the observed increase in cumulative mortality with greater sampling depth was minor: from 17 to 20%. However, the comparison of mortality after samplings from the same depth clearly shows higher rates for *O. albinus*: 4.4 times higher from 200 m (but those animals were sampled at different years) and 1.8 times higher from 300 m (the same trap). Unfortunately, we could not determine and freeze the amphipods right before or after death, and the following biochemical measurements were made only for the surviving part of the population.

### 3.5. Influence of Laboratory Acclimation on Biochemical Traits

Since we were able to perform the laboratory acclimation with amphipods from only some of the sampling depths, we decided to compare the biochemical traits of acclimated animals with non-acclimated individuals from the same traps only. The available data for energy-related parameters and oxidative status markers are displayed in Figure 5 and Figure 6, respectively.

Unfortunately, for most metabolic traits, the comparisons are only available for the first sampling campaign. Nevertheless, the comparison for glycogen demonstrated a statistically significant decrease in *O. flavus* after the acclimation (Figure 5), which effectively equalized the median level with the glycogen content in *O. albinus*. This follows up the increased glycogen level in *O. flavus* in the first sampling campaign mentioned earlier, which was probably related to accidentally elevated food availability for this species. The concurring increase in the ATP level and AEC in *O. flavus* after the acclimation (Figure 5) might be related to metabolic conversion of the excess glycogen.

However, the most notable difference among metabolic parameters was observed for LDH (Figure 5). The median activity of this enzyme rose in acclimated *O. flavus* by several times, and in both years, the difference was statistically significant. In the case of *O. albinus*, the increase in LDH activity was comparable in value though non-statistically significant.

The biochemical traits related to the cell oxidative status were analyzed in both years (Figure 6). In the first sampling campaign, we found a statistically significant increase in the CAT activity of acclimated *O. flavus* and similar increases in the GST activity of both studied species, but these observations were not repeated in the second sampling campaign. In the second sampling campaign, we also found a statistically significant doubling of POD activity for *O. albinus* after the acclimation, and concurrently, an approximately 1.5 times decrease in the DC content of neutral lipids. Thus, unlike the LDH increase, we observed no changes in the studied markers of oxidative stress that were consistent over two years.

## 4. Discussion

In this study, we unambiguously confirmed that an endemic amphipod from Lake Baikal, *O. albinus*, prefers greater depths than *O. flavus* and indeed may be of interest as an important freshwater model species adapted to high hydrostatic pressure and possessing other features of deep-water animals. The distinction in preferred pressures and depths may go in line with some other interspecific ecological differences (for example, in the sensitivity to higher temperatures and solar radiation) yet to be studied. Nevertheless, no less than 80% of *O. albinus* individuals survive and stay active after long-term acclimation to atmospheric pressure, despite the fact that the mortality of this species is higher than in the case of *O. flavus*. Importantly, both species fully survive short-term pressure decreases during lifting from the deep-water zone.

We concentrated our comparison of *O. flavus* and *O. albinus* on metabolic and oxidative parameters, since we expected distinct reactions in those markers, indicating stress tolerance due to the difference in preferred depths. In particular, we could expect a better energy supply for *O. flavus*, which would be indicated by a higher level of glycogen storage and ATP concentration and lower contents of glucose, ADP and AMP, as well as lower LDH activity [47,48]. Similarly, misregulation of the mitochondrial electron transport chain in *O. albinus* at low pressure could lead to elevated production of ROS and, therefore, to increased levels of lipid peroxidation products, as well as increased activity of antioxidant enzymes, which cope with oxidative stress [49,50].

In fact, the results obtained for these biochemical traits mostly highlight the similarities between the studied *Ommatogammarus* species. Right after sampling, we could not find consistent interspecific differences in any measured parameters except GST activity and DC content in the storage lipids. Long-term acclimation to laboratory conditions at atmospheric pressure led only to a consistent increase in the LDH activity of *O. flavus*. A comparable rise in median LDH levels was also observed for *O. albinus*, but these changes were not statistically significant, and thus, the described effect requires further research for this species.

If *O. albinus* experienced oxidative stress during and right after the pressure decrease, we would expect higher GST activity and a higher DC content in *O. albinus* than in *O. flavus*. However, we observe the opposite for both markers. We could not trace the analyzed traits in the individuals that died during long-term laboratory acclimation, but the surviving parts of populations of both species showed comparable values. Thus, we found no consistent biochemical evidence that the metabolism of *O. albinus* is less adapted to atmospheric pressure than the metabolism of *O. flavus* at the levels of energy supply and control for ROS production. Importantly, we found no influence of the sampling depth on any of the measured markers for both species, which indicates the good tolerance of these scavengers to the applied quick pressure decreases even from over 70 atm, at least for the short-term.

*O. albinus* has a higher amount of total lipids than *O. flavus* [51], and the main difference is probably associated with storage lipids, as the deeper-dwelling *O. albinus* should require larger energy storage (and it is seemingly “fatter”). This difference may explain the lower DC content in the storage lipids of *O. albinus*, since the same amount of produced DC may be normalized to a higher amount of lipids. Other lipid peroxidation products in storage lipids also have lower median values in *O. albinus* than in *O. flavus*, although the differences are mostly non-significant. The interspecific difference in GST activity is more difficult to explain. GST participates in eliminating various toxicants, not only products of oxidative stress [52]. The absence of a difference in CAT and POD activities suggests a demand of *O. flavus* for the detoxification of some other substances not related to oxidative stress. It is conceivable that *O. flavus* consumes a higher amount of some toxic substances in food than *O. albinus*, but the previous data on lipid composition suggest identical diets for these species [51].

The increase in LDH activity in *O. flavus* after long-term acclimation might indicate a higher rate of anaerobic energy production in glycolysis [23], probably due to elevated energy demand or decreased activity in the mitochondrial electron transport chain. It can thus be a sign of sub-optimal conditions even for the surviving individuals, which is not surprising for deep-water animals. We cannot yet argue that this effect is specific for *O. flavus* as this may be the case for *O. albinus* as well. However, this effect additionally demonstrates that hyperbaric chambers may be essential for some long-term laboratory experiments, even for the deep-water animals that are able to effectively tolerate atmospheric pressure in the short term.

Overall, our data demonstrate that *O. albinus* can be safely sampled from deep-water zones of Lake Baikal and even (at least for certain types of experiments) kept at atmospheric pressure without critical metabolic disturbances, along with *O. flavus*. Sampling a high number of *Ommatogammarus* amphipods from the stable ice cover during late winter and early spring is less troublesome in comparison to most of their marine counterparts, which makes these unique freshwater crustaceans very attractive for physiological research. In light of these facts, we believe that this pair of evolutionarily close but ecologically distinct species, *O. flavus* and *O. albinus*, will serve the field of environmental physiology well in better understanding the adaptive potential and specific features of deep-water crustaceans in general.

## Figures and Tables

**Figure 1 insects-13-00578-f001:**
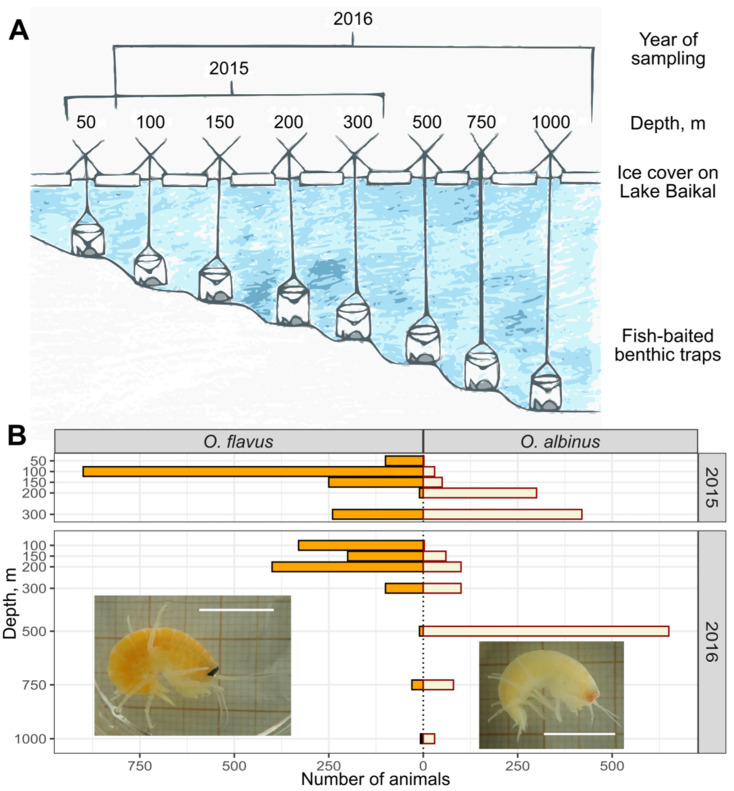
Sampling design and results. (**A**) Scheme of deep-water traps installed at the bottom of Lake Baikal in the first sampling campaign (2015) and the second sampling campaign (2016). The “X” signs represent mounts for the traps on the ice surface. (**B**) Approximate abundance of adult individuals of *O. flavus* and *O. albinus* in the traps, with respective photos of these species (the scale bar is 1 cm).

**Figure 2 insects-13-00578-f002:**
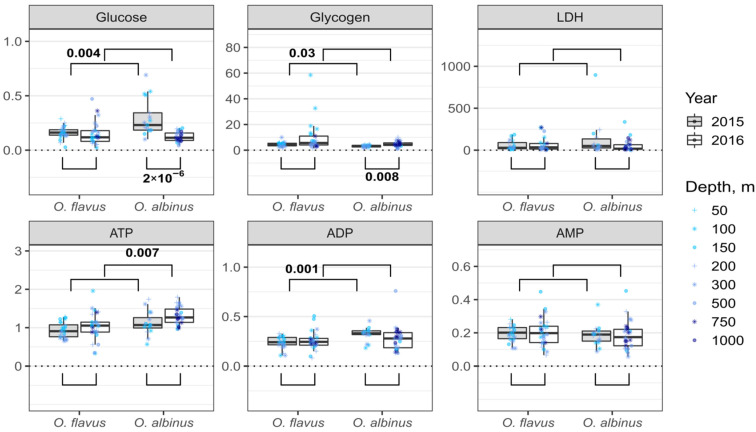
Energy-related biochemical parameters in *O. flavus* and *O. albinus* fixed right after collection from deep-water traps. Measurements for amphipods from all depths were pooled within each sampling year; *n* = 27–30 for *O. flavus* and *n* = 21–26 for *O. albinus*. Values are expressed as μmol g^−1^ of tissue for adenylates, glucose and glycogen (glucose-corrected) reserves, and as nkat mg^−1^ for LDH activity. The numbers near the brackets indicate *p*-values for respective pairwise comparisons if the difference is statistically significant (Mann–Whitney U-test with Holm’s correction for multiple comparisons). A similar comparison that only includes the samples from 100–300 m is presented in Appendix A.

**Figure 3 insects-13-00578-f003:**
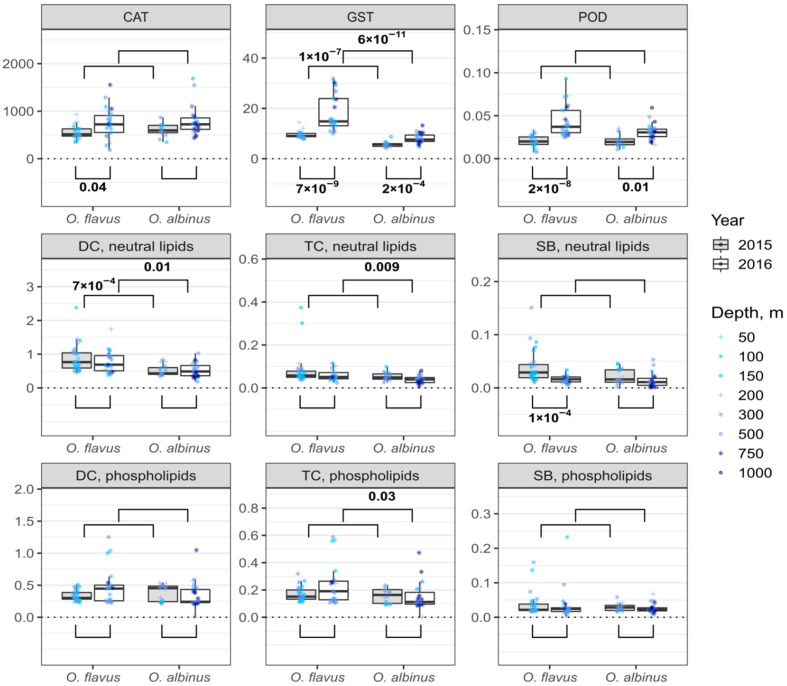
Biochemical parameters related to oxidative status in *O. flavus* and *O. albinus* fixed right after collection from deep-water traps. Measurements for amphipods from all depths were pooled within each sampling year; *n* = 27–30 for *O. flavus* and *n* = 21–26 for *O. albinus*. Values are expressed as nkat mg^−1^ of protein for the enzyme activities (CAT, GST, POD) and in a.u. for lipid peroxidation products (DC, TC and SB). The numbers near the brackets indicate *p*-values for respective pairwise comparisons if the difference is statistically significant (Mann–Whitney U-test with Holm’s correction for multiple comparisons). A similar comparison that only includes the samples from 100–300 m is presented in Appendix A.

**Figure 4 insects-13-00578-f004:**
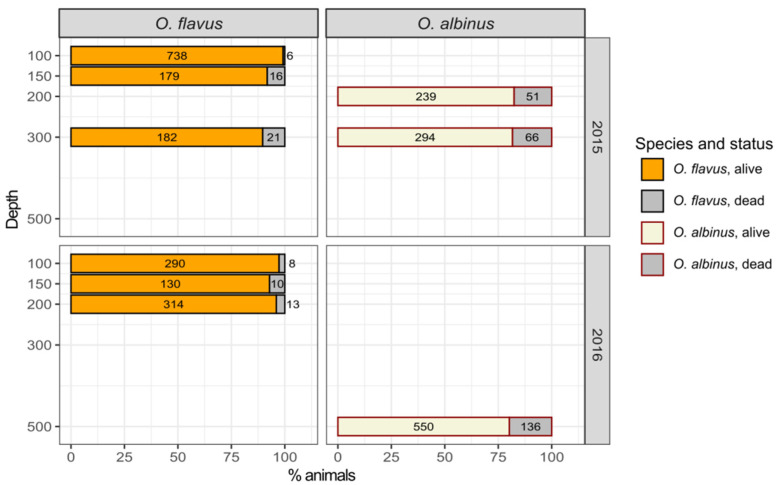
Survival of *O. flavus* and *O. albinus* after long-term acclimation to laboratory conditions at atmospheric pressure. In the 1st sampling campaign, the acclimation lasted 9–12 days for both species; in the 2nd sampling campaign, it lasted 18–27 days for *O. flavus* and 13–16 days for *O. albinus*.

**Figure 5 insects-13-00578-f005:**
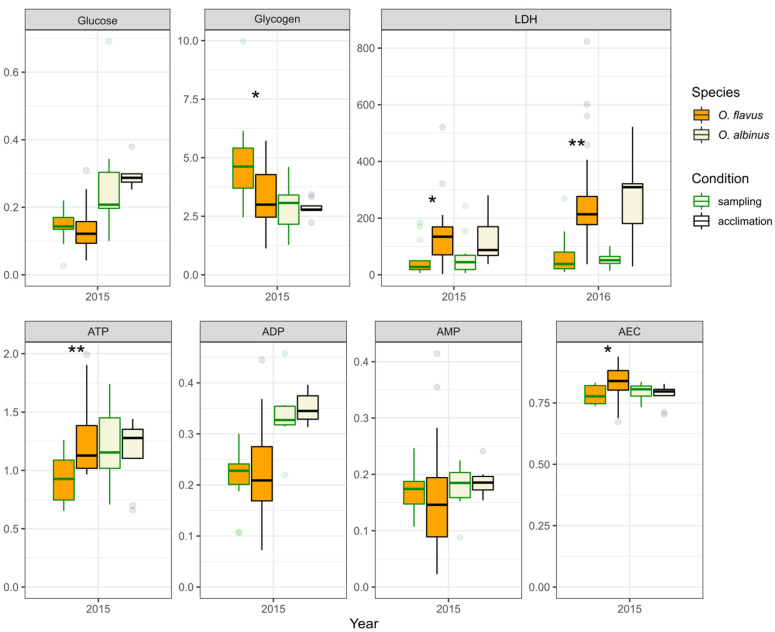
The influence of acclimation to atmospheric pressure on energy-related biochemical parameters in *O. flavus* and *O. albinus*. Data for amphipods fixed right after collection were subsetted to match the sampling depths with the groups after acclimation. *O. flavus* in the 1st sampling campaign: 100, 150 and 300 m; in the 2nd sampling campaign: 100, 150 and 200 m. *O. albinus* in the 1st sampling campaign: 200 and 300 m; in the 2nd sampling campaign: 500 m. In the 1st sampling campaign, the acclimation lasted 9–12 days for both species; in the 2nd sampling campaign, it lasted 18–27 days for *O. flavus* and 13–16 days for *O. albinus*. Values are expressed as μmol g^−1^ of tissue for adenylates, glucose and glycogen (glucose-corrected) reserves and as nkat mg^−1^ for LDH activity. *n* = 5–25; outlier data points beyond the ends of the whiskers are plotted individually. Asterisks indicate *p*-values within different ranges: *—*p* < 0.05; **—*p* < 0.01 (Mann–Whitney U-test with Holm’s correction for multiple comparisons).

**Figure 6 insects-13-00578-f006:**
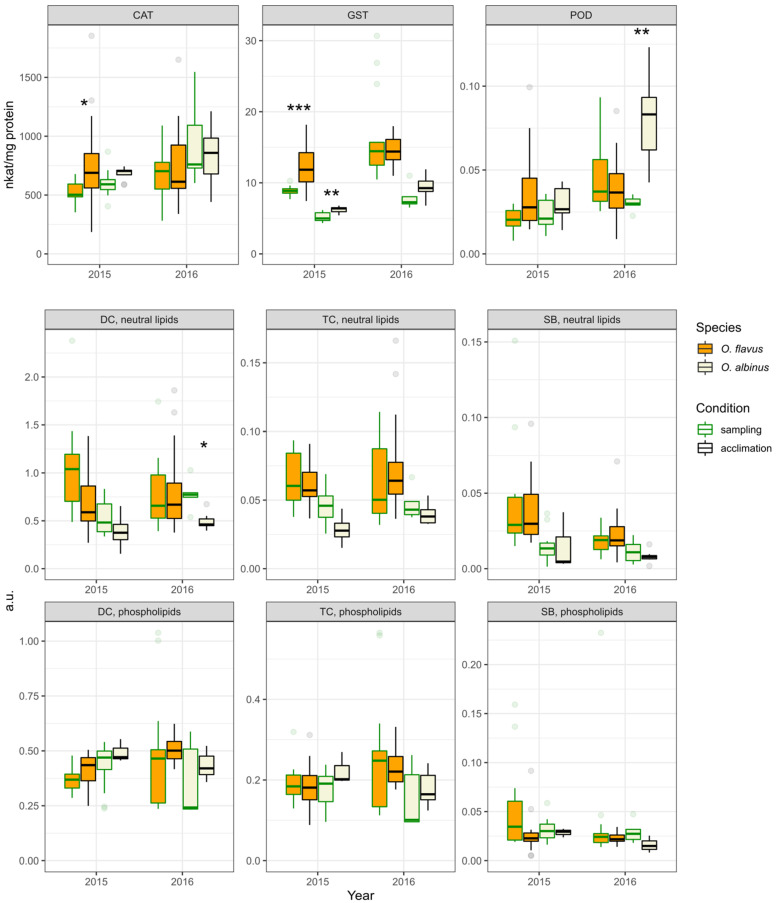
The influence of the acclimation to atmospheric pressure on biochemical parameters related to oxidative status in *O. flavus* and *O. albinus*. Data for amphipods fixed right after collection were subsetted to match the sampling depths with the groups after acclimation. *O. flavus* in the 1st sampling campaign: 100, 150 and 300 m (only 100 and 300 m for lipid peroxidation products); in the 2nd sampling campaign: 100, 150 and 200 m. *O. albinus* in the 1st sampling campaign: 200 and 300 m; in the 2nd sampling campaign: 500 m. In the 1st sampling campaign, the acclimation lasted 9–12 days for both species; in the 2nd sampling campaign, it lasted 18–27 days for *O. flavus* and 13–16 days for *O. albinus*. Values are expressed as nkat mg^−1^ of protein for the enzyme activities (CAT, GST, POD) and in a.u. for lipid peroxidation products (DC, TC, and SB). *n* = 4–35; outlier data points beyond the ends of the whiskers are plotted individually. Asterisks indicate *p*-values within different ranges: *—*p* < 0.05; **—*p* < 0.01; ***—*p* < 0.001 (Mann–Whitney U-test with Holm’s correction for multiple comparisons).

## Data Availability

All the obtained data are available in Appendix A.

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
