# Peer review of "Metabolic Tolerance to Atmospheric Pressure of Two Freshwater Endemic Amphipods Mostly Inhabiting the Deep-Water Zone of the Ancient Lake Baikal"

_insects, 2022, doi:10.3390/insects13070578_

Round 1
Reviewer 1 Report
Reviewer Comments to Author
General comments
The manuscript “Metabolic tolerance to atmospheric pressure of two freshwater endemic amphipods mostly inhabiting the deep-water zone of the ancient Lake Baikal” represents an exploration two species from an order that has been characterized by several species to be proposed as model species in ecotoxicological assessments, from freshwater ecosystems (such as the lake Baikal) to coastal and estuarine ecosystems. The present work represents well the adaptive abilities of two gammarid species to high pressure variations and their suitability to be used in research as model organisms, mostly regarding environmental physiology. The manuscript is very well written. I have just some concerns regarding the methodology and how the results are presented and therefore discussed. This is mentioned in detail in the specific comments and should be addressed.
Specific comments
The experimental design is not the most adequate to compare different species from different years by taking in consideration the different depth ranges used. I would compare only 2015 and 2016 considering the same ranges (100 to 300 m). Since most organisms from the O. albinus species were found at 500 meters in 2016, I would perform comparisons with this particular group separately, found deeper and at a higher pressure. The argument that both species resist to high pressure fluctuations and can be considered as suitable fresh water models for physiological assessments under laboratory conditions would be more clear and the data treated more correctly, in my opinion. Possibly the authors already performed this exercise and gave no differences but this needs to be stated in the manuscript, at least.
Figure 1A (representing the sampling design) should be presented in the methodology section 2.1, better than in the introduction. In the introductions section I would like to see more about the species in question, or the order amphipoda. Figure 1B represents the abundance found for both species and should be thus presented in the results section 3.1.
Line 121 – I would not say that the species were maintained in as dark conditions as possible, since the experiments were in fact conducted in dark conditions. Only for water changes they were removed from these permanent conditions, as mentioned in the manuscript.
Line 135 – The authors should be more specific about which parameters were evaluated with 2 and 3 analytical replicates as all measurements should be measured with at least three replicates, but which is more important for the enzymatic activities.
Line 151 – The authors adapted the methodology for content of lipid peroxidation products, which is a very interesting and differentiating approach to follow, not considering only simple malondialdehyde products. However, and given that it is not easy to access the works from where this methodology was adapted, I encourage the authors to give a more detailed description in this section.
Line 183 – The link provided by the authors does not lead to the set of scripts used.
Line 249 – In the graphics from figure 3 it is mentioned DC, TC. In the legend from the same figure it is mentioned SK and TK. Is this intentional? The same for figure 5.
Author Response
Dear Reviewer 1,
We would like to deeply thank you for your effort in helping us to improve the manuscript and for high estimate of our work. Please find our point-by-point response to your concerns below (as an attached file as well).
General comments
The manuscript “Metabolic tolerance to atmospheric pressure of two freshwater endemic amphipods mostly inhabiting the deep-water zone of the ancient Lake Baikal” represents an exploration two species from an order that has been characterized by several species to be proposed as model species in ecotoxicological assessments, from freshwater ecosystems (such as the lake Baikal) to coastal and estuarine ecosystems. The present work represents well the adaptive abilities of two gammarid species to high pressure variations and their suitability to be used in research as model organisms, mostly regarding environmental physiology. The manuscript is very well written. I have just some concerns regarding the methodology and how the results are presented and therefore discussed. This is mentioned in detail in the specific comments and should be addressed.
Specific comments
The experimental design is not the most adequate to compare different species from different years by taking in consideration the different depth ranges used. I would compare only 2015 and 2016 considering the same ranges (100 to 300 m). Since most organisms from the O. albinus species were found at 500 meters in 2016, I would perform comparisons with this particular group separately, found deeper and at a higher pressure. The argument that both species resist to high pressure fluctuations and can be considered as suitable fresh water models for physiological assessments under laboratory conditions would be more clear and the data treated more correctly, in my opinion. Possibly the authors already performed this exercise and gave no differences but this needs to be stated in the manuscript, at least.
MT: Indeed, we already tried to make comparisons in this and other ways. We previously prepared Figure S1 and also performed the correlation analysis specifically to formally demonstrate that measurements from the greater depths are in fact very similar to the measurements in the range 50-300 m for both species. However, in order to meet your (and probably other readers’) concern we now prepared new Figures S2 and S3 analogous to Figures 2 and 3 but only with data from 100-300 m. Respective description is of course now added to the Results section. We decided not preparing separate figure for comparing 100-300 m and 500-1000 m specifically for O. albinus from 2016 since current Figure S1 should well serve this purpose to our opinion.
In addition to the new text at the end of section 3.3 we would like to highlight that even within the range 100-300 m the vertical distribution of two species is substantially shifted. It is especially true for 2015, when O. flavus was predominantly sampled from 100 m and O. albinus from 300 m, but it is the case for 2016 as well. So, the pressure decrease is different for the majority of individuals of O. flavus and O. albinus by 2-3 times even for Figures S2 and S3. Thus, if we would like to make an interspecies comparison that would be absolutely unbiased by the difference in pressure decrease, we would have to keep just one depth for both years. It is possible for 300 m, but because of low group sizes it gives almost no statistically significant differences despite some medians are quite different (such as GST and DC in neutral lipids, which are significantly different between species in both Figure 3 and Figure 3S), please check below:
[available in the attached file]
Thus, we suppose new Figures S2 and S3 support the originally made conclusions rather than contradict them.
In the introductions section I would like to see more about the species in question, or the order amphipoda.
MT: We’ve added substantially more information about marine deep-water amphipods and a remark about convergently similar morphology of marine scavengers and the genus Ommatogammarus to the Introduction. We would like to thank you for this highly useful recommendation.
Figure 1A (representing the sampling design) should be presented in the methodology section 2.1, better than in the introduction… Figure 1B represents the abundance found for both species and should be thus presented in the results section 3.1.
MT: We totally agree that it seems more logical to place Fig 1A and Fig 1B in the Methods and Results respectively. However, the scheme of trap installation is very different between two years and not easy to remember. In our opinion and to our experience (as concluded from internal peer review), the current Fig. 1B would be very confusing for most readers without Fig. 1A placed just side-by-side. If they are separated, the difference between years on Fig. 1B could be erroneously understood as natural variability in the animal vertical distribution. Thus, in order to follow both motivations, we moved Fig. 1 to the Methods section.
Line 121 – I would not say that the species were maintained in as dark conditions as possible, since the experiments were in fact conducted in dark conditions. Only for water changes they were removed from these permanent conditions, as mentioned in the manuscript.
MT: We now tried to clarify this issue in the Methods section. The incubators we used have the central window (looks like this https://www.labmakelaar.eu/wp-content/uploads/2022/04/Sanyo-MIR-252-Refrigerated-Incubator_Koelbroedstoven_37811_5-scaled.jpeg), and unfortunately we didn’t cover them completely in order to check the aquaria from time to time. However, we kept the lights in room switched off and the incubators were located in the shadow from the room window. We’ve now checked the actual illuminance values with a lux meter and added them to the manuscript. We would like to thank you for pointing out this ambiguity.
Line 135 – The authors should be more specific about which parameters were evaluated with 2 and 3 analytical replicates as all measurements should be measured with at least three replicates, but which is more important for the enzymatic activities.
MT: We used 2 analytical replicates only for adenylates, glucose and glycogen. All other parameters were measured in 3 analytical replicates. This is now specified in the text.
Line 151 – The authors adapted the methodology for content of lipid peroxidation products, which is a very interesting and differentiating approach to follow, not considering only simple malondialdehyde products. However, and given that it is not easy to access the works from where this methodology was adapted, I encourage the authors to give a more detailed description in this section.
MT: We previously described the technique in detail in Vereshchagina et al., 2018, DOI: 10.7717/peerj.5571. The reference is now added to the Methods section along with a more extended description in the manuscript text.
Line 183 – The link provided by the authors does not lead to the set of scripts used.
MT: We’ve checked the link again and it works for us. We removed the point at the end of the sentence to make the link easier to copy.
Line 249 – In the graphics from figure 3 it is mentioned DC, TC. In the legend from the same figure it is mentioned SK and TK. Is this intentional? The same for figure 5.
MT: Indeed, these were our typos in the legends. We are grateful for the correction.
We would like to thank you again for your work.
Very sincerely yours,
Dr. Sci., Prof. Maxim A. Timofeyev
Irkutsk State University

Reviewer 2 Report
Main Summary:
The manuscript by Madyarova et al. characterizes some aspects of the metabolic phenotype of two species of amphipods collected from various depths in Lake Baikal. They seek to establish the viability of using these species as model species for the study of adaptation to deep water habitats. They represent an independent radiation into deep waters from their marine relatives; and the authors looked for signs of metabolic stress during collection and captivity. O. flavus is the more shallow of the two species, and has a more variable depth range (from surface waters to 1300 m), while O. albinus has a deeper depth range.
The authors measured various endpoints of oxidative stress and energy metabolism. O. albinus had lower survivorship than its congener during captivity. Additionally, O. flavus showed increased LDH following acclimation which indicates that the low pressure may be stressful over long periods of time. They concluded that both species could be used as model species for investigating the adaptation mechanisms of Baikal deep-water amphipods and comparing them to marine counterparts.
Overall the paper is well organized and the data are interesting. The authors can better explain their goals/hypotheses to improve the paper as described below.
Major Revisions:
The rationale for the paper is fairly clear, that these species could serve as model organisms for deep water adaptation studies. However, there is very little rationale for why the specific endpoints were chosen, nor information on why those endpoints would be good indicators of the utility of the species as the desired models. This information is only found in the discussion (which helps, starting on line 336), but even there it is not as fully developed as it should be. As a reader who is not a deep-water biologist, the metabolic endpoints are familiar, but their relationship to the key point of the paper is not clear. This especially needs to be put into the last paragraph of the introduction in detail and developed more in the discussion. The authors should also describe more information about what is known about deep water species and their metabolic response to changing pressures in the paragraph starting on line 50.
A more formal analysis of the survival would be interesting. I was surprised at the high survival shown. Perhaps these data are not sufficient for that question, but some more explanation of survival seems important as this is of course the most important thing for the species to be useful as the deep-water model. Addressing the data in S1 more candidly and directly in the text would help the reader understand the implications of the survivorship of the two species.
Minor Revisions:
The NS notation to denote non-significance is not necessary and can be removed– possibly making the figures simpler. This is a stylistic suggestion.
Throughout the paper, there are minor English errors that could be quickly corrected by a native speaker. These are distracting, but do not interfere with the flow of the paper, but should be corrected.
It is unclear if sampling is from the bottom (as suggested by figure 1) or if these are in the water column (as it seems from the text).
Figure 1, it says major size scale is 1 cm, but I do not see the scale bar anywhere.
Part A- the sampling schematic is unclear. The ice break and X's above the ice break are unclear to someone not-familiar with this sampling methodology. Either more explanation in the figure or in the legend would correct this.
Part B- A figure that shows the ranges of both species (in water depth) as collected over the two years would help the reader visualize the differences in the two species rather than four separate histograms. Perhaps the same axis for each species for 2015 and 2016 with abundances for each species on opposite sides of the y axis. (the y access being depth).
Figure 2: legend symbols need to be larger, no vertical lines within the actual graphs, and p values should be in bold or some other marker to draw attention.
The statement that they are ecologically distinct needs to be supported more (they share the same food, are both benthopalegic– what separates them ecologically? If nothing, remove this statement). (line 387)
Author Response
Dear Reviewer 2,
We would like to deeply thank you for your effort in helping us to improve the manuscript and for high estimate of our work. Please find our point-by-point response to your comments and suggestions below (as an attached file as well).
Main Summary:
The manuscript by Madyarova et al. characterizes some aspects of the metabolic phenotype of two species of amphipods collected from various depths in Lake Baikal. They seek to establish the viability of using these species as model species for the study of adaptation to deep water habitats. They represent an independent radiation into deep waters from their marine relatives; and the authors looked for signs of metabolic stress during collection and captivity. O. flavus is the more shallow of the two species, and has a more variable depth range (from surface waters to 1300 m), while O. albinus has a deeper depth range.
The authors measured various endpoints of oxidative stress and energy metabolism. O. albinus had lower survivorship than its congener during captivity. Additionally, O. flavus showed increased LDH following acclimation which indicates that the low pressure may be stressful over long periods of time. They concluded that both species could be used as model species for investigating the adaptation mechanisms of Baikal deep-water amphipods and comparing them to marine counterparts.
Overall the paper is well organized and the data are interesting. The authors can better explain their goals/hypotheses to improve the paper as described below.
Major Revisions:
The rationale for the paper is fairly clear, that these species could serve as model organisms for deep water adaptation studies. However, there is very little rationale for why the specific endpoints were chosen, nor information on why those endpoints would be good indicators of the utility of the species as the desired models. This information is only found in the discussion (which helps, starting on line 336), but even there it is not as fully developed as it should be. As a reader who is not a deep-water biologist, the metabolic endpoints are familiar, but their relationship to the key point of the paper is not clear. This especially needs to be put into the last paragraph of the introduction in detail and developed more in the discussion. The authors should also describe more information about what is known about deep water species and their metabolic response to changing pressures in the paragraph starting on line 50.
MT: Indeed, the storyline requires the suggested explanations. We added new relevant remarks to the second paragraph of the Introduction (again, mainly relying on the recent most relevant review by Yancey, 2020, DOI: 10.1002/jez.2354), substantially expanded the last paragraph of the Introduction in order to support the choice of markers and added new comments to the Discussion. We would like to thank you for the suggested corrections.
A more formal analysis of the survival would be interesting. I was surprised at the high survival shown. Perhaps these data are not sufficient for that question, but some more explanation of survival seems important as this is of course the most important thing for the species to be useful as the deep-water model. Addressing the data in S1 more candidly and directly in the text would help the reader understand the implications of the survivorship of the two species.
MT: We discussed this suggestion and decided that the survival data indeed deserve a separate figure in the main manuscript. Please find them now in the new Figure 4. We also found an error in formula in previous Table S1, which made the mortality percents for both species even slightly higher than they were in reality (now corrected both in Table S1 and Figure 4). Unfortunately, this comparison is pretty approximate due to the differences in acclimation times. We also don’t have the exact data for survival at different time points of the acclimation, so we see no means to compare two species with some other statistical tests rather than simple correlation with depth. We would like to thank you for this especially useful recommendation.
Minor Revisions:
The NS notation to denote non-significance is not necessary and can be removed– possibly making the figures simpler. This is a stylistic suggestion.
MT: It is now corrected for both Figures 2 and 3.
Throughout the paper, there are minor English errors that could be quickly corrected by a native speaker. These are distracting, but do not interfere with the flow of the paper, but should be corrected.
MT: The manuscript has been now proofread. We are grateful for this recommendation.
It is unclear if sampling is from the bottom (as suggested by figure 1) or if these are in the water column (as it seems from the text).
MT: The sampling was always from the lake bottom. Now it’s unambiguously stated in the Methods as well.
Figure 1, it says major size scale is 1 cm, but I do not see the scale bar anywhere
MT: The explicit scale bars are now added to the photos.
Part A- the sampling schematic is unclear. The ice break and X's above the ice break are unclear to someone not-familiar with this sampling methodology. Either more explanation in the figure or in the legend would correct this
MT: Xs depict the mounts for the trap ropes to the ice surface, as simple as that. Additional explanation is now added to the Fig. 1A caption.
Part B- A figure that shows the ranges of both species (in water depth) as collected over the two years would help the reader visualize the differences in the two species rather than four separate histograms. Perhaps the same axis for each species for 2015 and 2016 with abundances for each species on opposite sides of the y axis. (the y access being depth).
MT: Among researchers working with Baikal amphipods it’s widely assumed that the scavengers are able to migrate for quite long distance, including vertically, to reach the rotting fish. At least, we cannot discard such a possibility. In such a case it’s entirely possible that the installation of traps itself completely changes the vertical distribution of amphipods, and the number of animals in the traps doesn’t actually represent the real preferred distribution of both species (despite difference between two species is apparent). The installation depths in two sampling campaigns were very different, and we believe that mixing the distribution abundances obtained in the two years within the same axes would be completely inappropriate. The distribution data from the second sampling campaign is probably close to the native distribution, but we still cannot be sure and would thus like to present the raw data as they are.
However, we totally agree that a convenient and clear comparison of two species is important, and in order to achieve it we indeed plotted the abundances on opposite sides of the y axis. We hope the updated Fig. 1B gives an easier side-by-side comparison of the species within the same axes separately for each year of sampling.
Figure 2: legend symbols need to be larger, no vertical lines within the actual graphs, and p values should be in bold or some other marker to draw attention.
MT: The legend size, fontface for the p-values are now corrected for both Fig. 2 and Fig. 3.
The statement that they are ecologically distinct needs to be supported more (they share the same food, are both benthopalegic– what separates them ecologically? If nothing, remove this statement). (line 387)
MT: The studied species certainly differ in preferred depths and, hence, preferred pressure ranges. We suppose this is also an ecological difference, which can be important during studying their mechanisms of adaptation to changing pressure. In order to better explain our point we added a new sentence (currently, the second one) to the beginning of Discussion. Additionally, our unpublished data suggest that O. albinus is indeed somewhat more sensitive to high temperature than O. flavus and certainly more sensitive to solar UV radiation due to lower carotenoid content.
We would like to thank you again for your work.
Very sincerely yours,
Dr. Sci., Prof. Maxim A. Timofeyev
Irkutsk State University

Round 2
Reviewer 1 Report
The authors have very clearly addressed the concerns raised by the reviewer, particularly those related with the experimental design. From my part, I now consider the maunscript suitable for publication in Insects journal.
Best regards